# Ultrasound Radiomics Nomogram Integrating Three-Dimensional Features Based on Carotid Plaques to Evaluate Coronary Artery Disease

**DOI:** 10.3390/diagnostics12020256

**Published:** 2022-01-20

**Authors:** Xiaoting Wang, Peng Luo, Huaan Du, Shiyu Li, Yi Wang, Xun Guo, Li Wan, Binyi Zhao, Jianli Ren

**Affiliations:** 1Department of Ultrasound, The Second Affiliated Hospital of Chongqing Medical University, Chongqing 400010, China; wxt19960511@163.com (X.W.); luopengjjyy@163.com (P.L.); lishiyu20220101@163.com (S.L.); wangyiglory@163.com (Y.W.); gx961125@163.com (X.G.); wanli13635359967@163.com (L.W.); 2Department of Cardiology, The Second Affiliated Hospital of Chongqing Medical University, Chongqing 400010, China; duhuaan20@126.com (H.D.); zby339444192@163.com (B.Z.)

**Keywords:** SYNTAX, radiomics, ultrasound image, carotid plaques, three-dimensional ultrasound

## Abstract

This study aimed to explore the feasibility of ultrasound radiomics analysis before invasive coronary angiography (ICA) for evaluating the severity of coronary artery disease (CAD) quantified by the SYNTAX score (SS). This study included 105 carotid plaques from 105 patients (64 low-SS patients, 41 intermediate-high-SS patients). The clinical characteristics and three-dimensional ultrasound (3D-US) features before ICA were assessed. Ultrasound images of carotid plaques were used for radiomics analysis. Least absolute shrinkage and selection operator (LASSO) regression, which generated several nonzero coefficients, was used to select features that could predict intermediate-high SS. Based on those coefficients, the radiomics score (Rad-score) was calculated. The selected clinical characteristics, 3D-US features, and Rad-score were finally integrated into a radiomics nomogram. Among the clinical characteristics and 3D-US features, high-density lipoprotein (HDL), apolipoprotein B (Apo B), and plaque volume were identified as predictors for distinguishing between low SS and intermediate-high SS. During the radiomics process, 8 optimal radiomics features most capable of identifying intermediate-high SS were selected from 851 candidate radiomics features. The differences in Rad-score between the training and the validation set were significant (*p* = 0.016 and 0.006). The radiomics nomogram integrating HDL, Apo B, plaque volume, and Rad-score showed excellent results in the training set (AUC, 0.741 (95% confidence interval (CI): 0.646–0.835)) and validation set (AUC, 0.939 (95% CI: 0.860–1.000)), with good calibration (mean absolute errors of 0.028 and 0.059 in training and validation sets, respectively). Decision curve analysis showed that the radiomics nomogram could identify patients who could obtain the most benefit. We concluded that the radiomics nomogram based on carotid plaque ultrasound has favorable value for the noninvasive prediction of intermediate-high SS. This radiomics nomogram has potential value for the risk stratification of CAD before ICA and provides clinicians with a noninvasive diagnostic tool.

## 1. Introduction

As the main cause of death in developed and developing countries, coronary artery disease (CAD), which is the narrowing or blocking of the lumen of vessels, has brought considerable economic and health burdens to the global population. In the management process of patients with CAD, identifying the severity of coronary artery narrowing and consequently selecting the appropriate treatment measures are essential [1]. Invasive coronary angiography (ICA) is currently the reference standard for the diagnosis of CAD in patients with suspected coronary artery dysfunction; it can not only clarify the presence or absence of coronary artery stenosis but also determine the anatomy and characteristics of the coronary lesion, such as the location, degree, and extent of stenosis [2]. This invasive process, however, has many negative effects on patients, such as high radiation exposure, considerable costs, and formation of pseudoaneurysms. In addition, ICA is not suitable for some patients in special situations. ICA and subsequent intervention are not routine processes for patients with acute coronary syndromes who suffer from chronic kidney disease [3]. Although computed tomographic coronary angiography can eliminate obstructive CAD before ICA and consequently reduce unnecessary ICAs [4], it can still expose patients to a certain degree of radiation. Therefore, in clinical practice, there is a lack of a completely noninvasive tool to assess CAD.

Atherosclerosis is a leading cause of CAD. Because atherosclerosis is a systemic disease, there is a relationship between carotid artery atherosclerosis and coronary atherosclerosis [5]. Therefore, CAD may be assessed by evaluating carotid plaques. Radiomics, an emerging medical image analysis tool, extracts a large number of quantitative features from regions of interest (ROIs) in medical images for clinical analysis. Early radiomics studies showed that it has great promise for tumor detection, diagnosis, and prognostic assessment [6,7,8,9]. Radiomics is currently used in carotid plaque studies focusing on the identification of plaque vulnerability. Radiographic and ultrasound-based texture analyses have been used to identify symptomatic carotid plaques with good results [10,11,12]. Radiomics is the extraction of internal information from ultrasound images of carotid plaques at the microscopic level invisible to the naked eye. Current clinical studies on the correlation between carotid plaque and coronary events mainly focus on the histopathological level. Therefore, this study aimed to provide new information on the correlation between carotid plaque and coronary events. Based on the theory that changes in gene expression due to gene mutations lead to alterations in medical images of the corresponding sites, radiomics holds promise. Since carotid plaque is formed by the gradual deposition of lipids in the intima, it is assumed that lipid deposition leads to changes in the corresponding medical images. Since atherosclerosis is general and systematic [5], it is assumed that radiomics based on carotid plaque can be used to evaluate CAD. In addition, three-dimensional ultrasound (3D-US) is now able to assess carotid plaque with considerable accuracy and reduce human measurement error in continuous monitoring [13,14], so it can possibly be used in the assessment of CAD.

Therefore, we sought to explore the feasibility of the noninvasive tool of 3D-US and ultrasound radiomics based on carotid plaque to predict the risk of CAD.

## 2. Methods

### 2.1. Study Population

The study was approved by the Ethics Committee of the Second Affiliated Hospital of Chongqing Medical University, which waived the requirement to obtain informed consent from the patients because all data and images in this study were anonymous. All data were obtained from the patients’ medical history. The inclusion criteria were as follows: ① patients with angina who underwent ICA and were clinically diagnosed with CAD from January 2021 to October 2021 and ② patients who underwent two-dimensional ultrasound (2D-US) and 3D-US examinations of the carotid artery and had a carotid plaque detected within 24 h of ICA. A total of 120 patients with CAD were included. The exclusion criteria were as follows: ① patients whose identification of the internal structure of the plaque was affected by the posterior acoustic shadow of the hyperechogenic plaques of the carotid artery (N = 2); ② patients whose 3D-US resolution of the extremely hypoechoic plaque was insufficient to discern the boundary of the plaque, which affected the extraction of the corresponding 3D-US features (N = 6); ③ patients whose outlines of the plaques were affected by thick subcutaneous soft tissue in the neck, which affected the display of the plaque (N = 5); ④ patients who underwent previous coronary artery stenting (N = 2). Ultimately, this study included 105 carotid plaques from 105 patients all included in the training set (73 males, mean age 63.4 years, age range 39–82 years; 32 females, mean age 69.0 years, age range 48–88 years). The verification set consisted of 35 patients randomly selected from these 105 patients (25 men, mean age 63.8 years, age range 45–82 years; 10 females, mean age 70.6 years, age range 48–81 years) (Figure 1).

Baseline clinical data were obtained from patient hospital records, including sex, age, low-density lipoprotein (LDL), triglycerides (TG), total cholesterol (TC), high-density lipoprotein (HDL), non-HDL, hypersensitive C-reactive protein (HS-CRP), hypertension status, systolic blood pressure, diastolic blood pressure, smoking status, diabetes status, body mass index (BMI), hemoglobin A1c (HbA1c), estimated glomerular filtration rate (eGFR), apolipoprotein B (Apo B), lipoprotein(a) (Lp(a)), and blood glucose.

### 2.2. Coronary Atherosclerosis Risk Stratification

The gold standard in this study was the SYNTAX score (SS) of ICA [15], which was calculated by two cardiovascular physicians with more than 3 years of experience in ICA, who were blinded to the other records. Any differences were settled through discussion. This score was a point system for risk stratification based on anatomical features and lesion characteristics (lesion location, severity, bifurcation, calcification, etc.) dependent on the coronary artery. It was used to target left main artery lesions and/or three-branch lesions, assess their complexity, and provide a preliminary judgment for surgical modality selection based on the level of the score, thus predicting major cardiovascular events in patients undergoing percutaneous coronary intervention (PCI). The specific classification and treatment principles were as follows: ① patients with complex multivessel disease and SS ≥ 33 points were more suitable for coronary artery bypass grafting (CABG) than PCI; ② patients with moderate left main artery disease (SS, 23–32) were recommended to undergo PCI; ③ patients with low-risk left main artery and three-vessel lesions with SS ≤ 22 were recommended to undergo PCI, which was comparable to CABG. In our study, the low-risk group (SS ≤ 22) included 64 patients, and the intermediate-high-risk group (SS > 22) contained 41 patients.

### 2.3. Imaging Acquisition and Carotid Plaque 3D-US Feature Extraction

Carotid artery ultrasound was performed in all patients within 24 h of ICA. High-resolution carotid plaque 3D-US imaging was performed by a carotid ultrasonographer with more than 3 years of experience in carotid artery ultrasound examination who was blinded to the ICA results, operating the PHILIPS EPIQ5 system (Philips Healthcare, Eindhoven, The Netherlands) equipped with a VL13-5 probe of 5–13 MHz. To ensure as much consistency as possible in the ultrasound images of each patient, we proposed adopting the criterion that the ultrasound beam was perpendicular to the vessel wall, adjusting the 2D gain and imaging when the carotid vessel lumen was echogenic. When the patient had multiple carotid plaques at the same time, we proposed including and analyzing the largest plaque. All images were imported into the offline VPQ mode of QLAB software, and 3D-US features were extracted, including plaque volume, maximum area reduction rate, normalized wall index, and grayscale median. We selected the keyframe when lumen thinning was most obvious and calculated the keyframe plaque thickness, plaque area, mean, median, and standard deviation.

### 2.4. Carotid Plaque Ultrasound Radiomics Feature Extraction, Dimension Reduction, and Radiomics Score

The radiomics process was shown in Figure 2. The outlining of ROIs of carotid plaques was performed manually. We imported the entire carotid plaque ultrasound DICOM data into the open-source software 3D-Slicer (version 4.13.0, https://www.slicer.org/ (accessed on 10 September 2021). The outlining of the carotid plaque border was performed by an ultrasonographer with more than 5 years of experience in carotid artery ultrasonography examination who was blinded to the ICA results, according to the border between the carotid plaque and echogenic carotid blood flow. After completing ROIs outlining, we extracted 851 features from each ROI. Least absolute shrinkage and selection operator (LASSO) regression was used to select features that could distinguish between low and intermediate-high SS. During LASSO, 10-fold cross-validation was used to select the tuning parameter (*λ*), and several nonzero coefficients were generated and used to calculate the radiomics score (Rad-score).

### 2.5. Models

In this study, models for predicting the risk stratification of coronary lesions were developed based on clinical characteristics (model A), 3D-US and ultrasound radiomics (model B), and a combination of clinical characteristics, 3D-US and ultrasound radiomics (model C). Binary logistic regression with backward stepwise selection was used to build the models. The sensitivity, specificity, accuracy, Youden’s index, and area under the receiver operating characteristic (ROC) curve (AUC) were used to quantify the performance of the models.

Validation set data were used to validate the efficacy of the models. Decision curve analysis (DCA) was performed to select the model that provided the most benefit to the patient (optimal model). Finally, a nomogram was constructed based on the optimal model.

### 2.6. Statistical Analysis

All statistical analyses were performed with SPSS Statistics 25.0 (IBM, Armonk, NY, USA) and R software (http://www.R-project.org (accessed on 10 October 2021)). Continuous variables in accordance with normal distribution were expressed as mean ± standard deviations, whereas continuous variables that were not normally distributed were expressed as medians and quartiles. Differences in continuous variables that were normally distributed between two groups were compared using *t*-tests, and continuous variables that were not normally distributed were compared using Mann–Whitney U tests. Differences in categorical variables were compared using chi-square tests. Univariate and multivariate analyses were used to identify clinical characteristics and carotid plaque 3D-US features that could predict intermediate-high SS. Variables with *p* < 0.05 in the univariate analysis were entered into the multivariate analysis. DeLong’s test was used to test whether the differences between different ROC curves were significant. A *p* < 0.05 indicated a statistically significant difference.

## 3. Results

### 3.1. Patient Clinical Characteristics

The clinical characteristics of all patients were summarized in Table 1. This study included 64 patients (61%) with low SSs and 41 patients (39%) with intermediate-high SSs.

### 3.2. Clinical Characteristic Selection

By using univariate and multivariate logistic regression analyses, we selected two features (HDL and Apo B) from the clinical features that had the ability to predict the characteristics of CAD risk stratification (Table 2).

### 3.3. Radiomics Score and 3D-US Characteristics

#### 3.3.1. Screening for Ultrasound Radiomics Features and Radiomics Scores

The optimal *λ*.min (log) [value of *λ* that gave minimum mean cross-validated error] = 0.0914 (−2.3925) was selected with 10-fold cross-validation. From the 851 features, LASSO eventually selected 8 features that were most capable of predicting intermediate-high risk of coronary atherosclerosis (original-firstorder-Minimum, original-glszm-SizeZoneNonUniformity, wavelet-LHL-firstorder-Skewness, wavelet-LHL-glszm-GrayLevelNonUniformity, wavelet-LHH-glszm-GrayLevelNonUniformityNormalized, wavelet-HHL-firstorder-Median, wavelet-LLL-firstorder-Skewness, and wavelet-LLL-glszm-SizeZoneNonUniformity) (Figure 3). The Rad-score was calculated as follows:Rad-score = 1.28112 × original-firstorder-Minimum + 0.00085 × original-glszm-SizeZoneNonUniformity + 0.69417 × wavelet-LHL-firstorder-Skewness + 0.00393 × wavelet-LHL-glszm-GrayLevelNonUniformity + 0.21954 × wavelet-LHH-glszm-GrayLevelNonUniformityNormalized − 25.07312 × wavelet-HHL-firstorder-Median + 0.2944 × wavelet-LLL-firstorder-Skewness + 0.00088 × wavelet-LLL-glszm-SizeZoneNonUniformity.

The difference in radiomics scores between low SS and intermediate-high SS patients was significant (Mann–Whitney *U* test, *p* = 0.016). A similar result was also found for the verification set (Mann–Whitney *U* test, *p* = 0.006).

#### 3.3.2. 3D-US Feature Selection

Univariate logistic regression analysis indicated that only plaque volume was an independent predictor for intermediate-high SS. After the process of feature selection, we selected one feature (plaque volume) (*p* = 0.014, Table 3) that had the ability to distinguish between low SS and intermediate-high SS.

### 3.4. Models Construction

We built three models. The first model was a clinical model (model A, HDL + Apo B) with an AUC of 0.648 (95% confidence interval [CI], 0.543–0.753). The sensitivity, specificity, accuracy, and Yorden index were 31.7%, 84.4%, 63.8%, and 0.161, respectively, in the training set. In the validation set, an AUC of 0.667 (95% CI, 0.485–0.848), a sensitivity of 42.9%, a specificity of 76.2%, an accuracy of 62.9%, and a Youden index of 0.191 were found. We then combined the carotid plaque 3D-US feature (plaque volume) and the radiomics feature (Rad-score), which were classified as ultrasound features, so the second model was an ultrasound model (model B, plaque volume + Rad-score) which yielded an AUC of 0.723 (95% CI, 0.627–0.818), a sensitivity of 24.4%, a specificity of 93.8%, an accuracy of 66.7%, and a Youden index of 0.182, which was validated in the validation set. The third model was a combined model (model C, HDL + Apo B + plaque volume + Rad-score) whose AUC, sensitivity, specificity, accuracy, and Youden index values were 0.741 (95% CI, 0.646–0.835) and 0.939 (95% CI, 0.860–1.000), 41.5% and 85.7%, 85.9% and 85.7%, 68.6% and 85.7%, 0.274 and 0.714 in the training and validation sets, respectively (Table 4 and Figure 4). Therefore, the predictive performance of Model C was significantly higher than that of Model A (AUC, 0.714 vs. 0.648 and 0.939 vs. 0.667 in training and validation sets, respectively) (DeLong’s test, *p* = 0.011 and 0.005 in the training and validation sets, respectively).

### 3.5. DCA

We performed DCA to evaluate the clinical usefulness of the models. The DCA curves showed that the combined model (HDL + Apo B + plaque volume + Rad-score) could help patients obtain the most benefit (Figure 5).

### 3.6. Development of a Nomogram

We integrated all the independent risk factors, including HDL, Apo B, plaque volume, and Rad-score, for predicting intermediate-high SS into a nomogram. A calibration graph showed favorable results, with mean absolute errors of 0.028 and 0.059 in the training and validation sets, respectively (Figure 6).

## 4. Discussions

In this study, we attempted to explore the relationship between carotid plaque and coronary artery severity using a radiomics approach. Finally, combining carotid plaque ultrasound-based radiomics, 3D-US, HDL, and Apo B, we developed a radiomics model to predict the severity of CAD. This model initially demonstrated valuable discriminatory ability, providing a noninvasive method to assess coronary severity before coronary angiography and providing clinicians with an advanced prediction of prognosis.

Low or intermediate-high SS reflects the number, location, and extent of coronary atherosclerosis [15]. Intermediate-high SS indicates that it is likely that the patient’s coronary arteries are more severely atherosclerotic, which has a greater impact on the patient’s life and quality of life, indicating a poor prognosis and a high likelihood of future cardiovascular events that may lead to life-threatening conditions. PCI will also treat CAD with different levels of SSs differently. In reality, the positive impact of accurately identifying such patients is enormous, both for clinical workup and for maximizing patient benefit. The use of noninvasive methods to distinguish such patients is particularly important and convenient when considering with the limitations of ICA.

Radiomics emerged from the study of tumors to evaluate the internal characteristics of lesions and to find information to approach many medical problems, such as tumor diagnosis, prognosis, and response to treatment [16]. Moreover, medical images contain a large amount of information, both visible and invisible to the naked eye. We can assume that as long as the lesion can be imaged medically, we can extract high-throughput data from it and analyze it to solve clinical problems [17]. In recent years, excellent work in radiomics has been performed in non-oncology research areas such as hemorrhage [18,19], infected stones [20], and liver fibrosis [21]. Given that atherosclerosis is a systemic change and coronary atherosclerosis is homologous to carotid atherosclerosis, we hypothesized that as carotid atherosclerosis progresses, the internal properties of carotid plaques change, which could reflect the degree of coronary atherosclerosis. Changes in the internal characteristics of carotid plaques bring about changes in the nature of ultrasound images, so ultrasound imaging based on carotid plaques through radiomics analysis has potential for predicting the degree of coronary atherosclerosis. Finally, as we expected, the results confirmed this hypothesis. Our preliminary exploration will help advance the study of radiomics in coronary atherosclerosis and provide a novel, noninvasive tool to clinically assess the severity of CAD in patients with chest pain.

In the present study, we found that radiomics features extracted from carotid plaque ultrasound images had the potential to predict coronary artery severity. We identified eight ultrasound radiomics features with the most potential to predict coronary artery severity: original-firstorder-Minimum, original-glszm-SizeZoneNonUniformity, wavelet-LHL-firstorder-Skewness, wavelet-LHL-glszm-GrayLevelNonUniformity, wavelet-LHH-glszm-GrayLevelNonUniformityNormalized, wavelet-HHL-firstorder-Median, wavelet-LLL-firstorder-Skewness, and wavelet-LLL-glszm-SizeZoneNonUniformity. These features were used to establish a radiomics score, which had excellent performance (AUC, 0.741). In addition, we found that one of the 3D-US features of carotid plaques (plaque volume) had the ability to identify patients with intermediate-high SSs. Previous studies had found that there was a correlation between the ultrasonic features of carotid plaques and SS. Studies have shown that plaque volume measurement is a screening tool for cardiovascular risk stratification [5,22], and our study drew a similar conclusion that plaque volume measurement is of great value in evaluating the prognosis of patients. Nobutaka Ikeda et al. [23] found that ultrasound-based carotid plaque scores, which were external measures of carotid plaques, had predictive value for SS. Unlike Nobutaka Ikeda’s study, our present study took quantitative data extracted from the automated 3D-US of carotid plaque without artificial measurements and found that plaque volume measurement might be a potential predictor for intermediate-high SS, which complemented the findings of the study on the association between ultrasound characteristics of carotid plaque and SS content. In addition to the ultrasound features of carotid plaques, this study found that HDL was a protective factor for coronary artery disease (OR, 0.21) and that Apo B was an independent risk factor for intermediate-high SS (OR, 4.6). That is, patients with angina pectoris with lower HDL and higher Apo B were more likely to have more severe coronary atherosclerosis, which was similar to the findings of previous studies [24,25,26,27] that abnormal lipid metabolism has a very important influence on the development of coronary atherosclerosis [28]. Previous studies had shown that radiomics had good performance in the assessment of coronary lesions, such as the identification of advanced atherosclerotic lesions [29], the assessment of coronary inflammation [30], and the prediction of coronary artery calcification and stenosis [31]. However, these studies had some limitations. They were based on computed tomography, which undoubtedly exposed the patient to certain radiation. In contrast, our study was based on ultrasound images of carotid plaques, which were free of radiological hazards and had protective value for patients. As we expected, the potential value of radiomics in predicting intermediate-high SS (AUC, 0.741; 95% CI: 0.646–0.835) in combination with clinical features (HDL and Apo B) and 3D-US features of carotid plaques (plaque volume) enriched the study of the association of carotid plaque with coronary atherosclerosis and provided a noninvasive method to assess coronary artery severity before coronary angiography.

Of course, this study had several limitations. First, as a preliminary study exploring the relation of carotid plaque radiomics and the severity of coronary atherosclerosis, the number of patients included in this study was small, and the general applicability of its findings need to be confirmed in real-world patients. Furthermore, all carotid plaque ultrasound images used in this study were guaranteed by us to be generated when the blood was echogenic, but because the tissue thickness and nature of the superficial carotid artery surface differed in each patient, the ultrasound beam energy reaching the carotid plaque also differed, which seemed to lead to challenges in the consistency of the imaging parameters. Therefore, imaging criteria such as CT and MRI are needed to determine the homogeneity of the images.

## 5. Conclusions

In conclusion, we developed a carotid plaque-based ultrasound radiomics nomogram for the noninvasive prediction of intermediate-high SS. This radiomics nomogram has potential value for the risk stratification of CAD before ICA and provides clinicians with a noninvasive diagnostic tool.

## Figures and Tables

**Figure 1 diagnostics-12-00256-f001:**
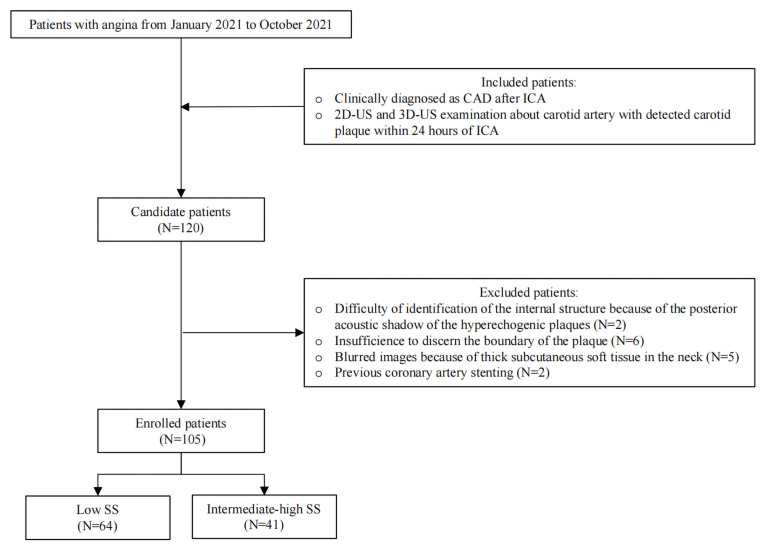
Flow chart of patient enrollment. CAD, coronary artery disease; ICA, invasive coronary angiography; 2D-US, two-dimensional ultrasound; 3D-US, three-dimensional ultrasound; SS, SYNTAX score.

**Figure 2 diagnostics-12-00256-f002:**
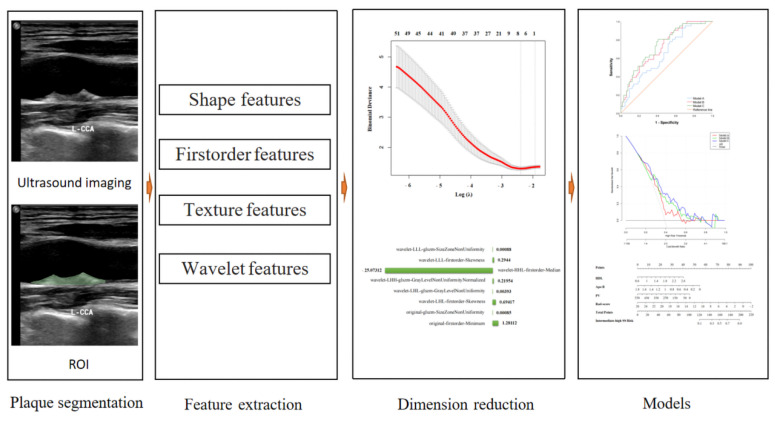
Workflow of the radiomics analysis to distinguish between low and intermediate-high SYNTAX scores.

**Figure 3 diagnostics-12-00256-f003:**
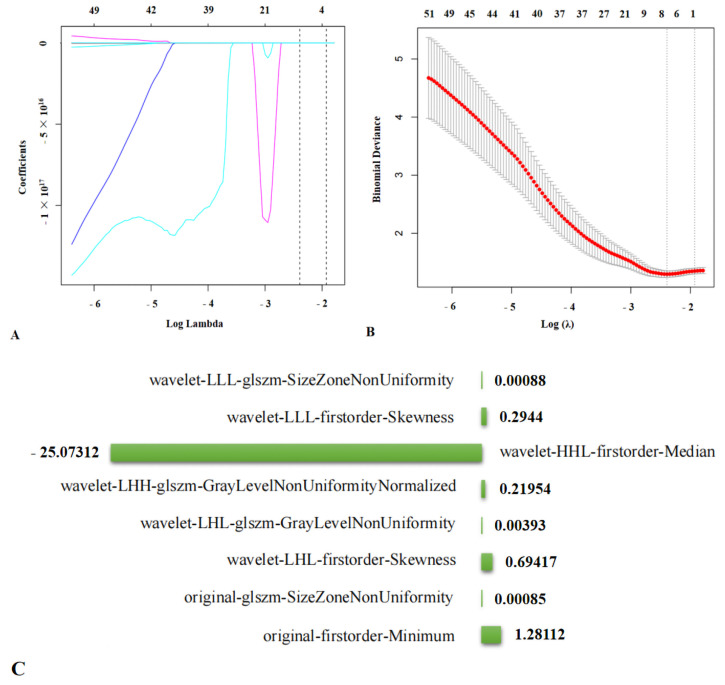
Feature selection process of radiomics in predicting intermediate-high SS. (**A**,**B**) An optimal tuning parameter (*λ*) value of 0.0914, with log(*λ*) = −2.3925 was determined by ten-fold cross-validation via minimum mean cross-validated error in LASSO model. Eight features that were most capable of predicting intermediate-high risk of coronary atherosclerosis were selected from 851 features. (**C**) The coefficients of the 8 features.

**Figure 4 diagnostics-12-00256-f004:**
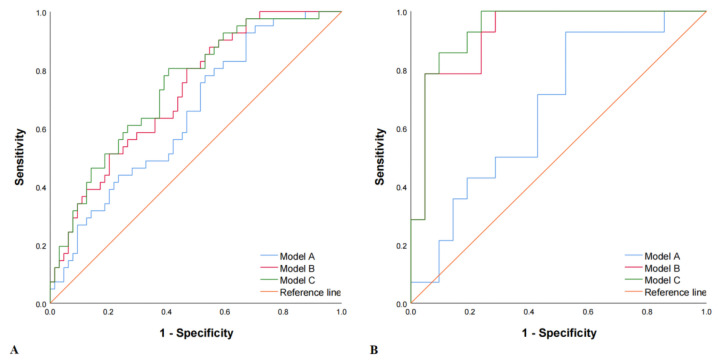
Receiver operating characteristic (ROC) curves in both the training and the validation set. Method A, Method B, and Method C represented the modeling methods of clinical features (HDL + Apo B), ultrasound features (plaque volume + Rad-score), and the combined model (HDL + Apo B + plaque volume + Rad-score). (**A**) Training set: the AUCs in Model A, Model B, and Model C were 0.648 (95% CI, 0.543–0.753), 0.723 (95% CI, 0.627–0.818) and 0.741 (95% CI, 0.646–0.835), respectively. (**B**) Validation set: the AUCs in Model A, Model B, and Model C were 0.667 (95% CI, 0.485–0.848), 0.922 (95% CI, 0.833–1.000) and 0.939 (95% CI, 0.860–1.000), respectively.

**Figure 5 diagnostics-12-00256-f005:**
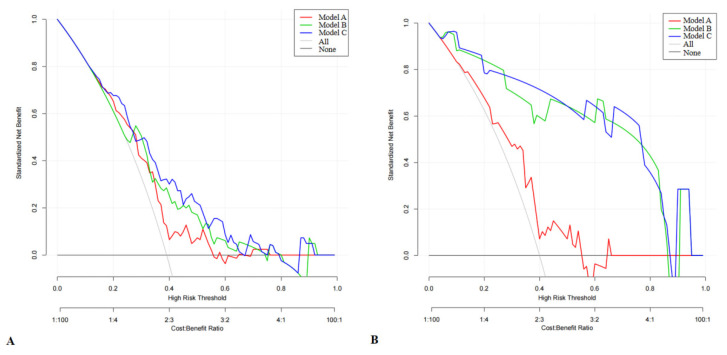
Decision curve analysis (DCA) of the three models in predicting intermediate-high SS. (**A**) Training set. (**B**) Validation set.

**Figure 6 diagnostics-12-00256-f006:**
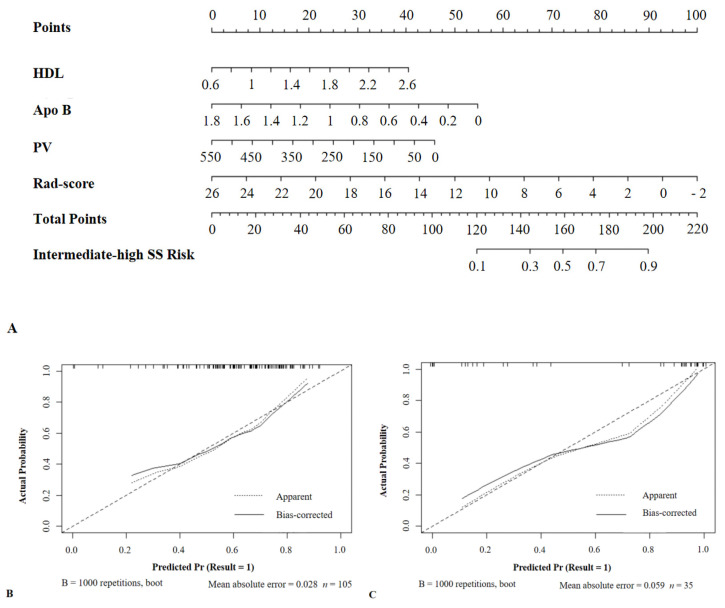
Nomogram combining HDL, Apo B, plaque volume, and Rad-score. (**A**) Nomogram plot. (**B**) Calibration graph for the training set. (**C**) Calibration graph for the validation set. HDL, high-density lipoprotein; ApoB, apolipoprotein B; PV, plaque volume; SS, SYNTAX scores. Result = 1: The SS was intermediate-high.

**Table 1 diagnostics-12-00256-t001:** Clinical characteristics of all patients.

Clinical Features	Low SSs (64)	Intermediate-High SSs (41)	*p* Value
Gender			0.051
Man	40 (38.1%)	33 (31.6%)	
Female	24 (22.9%)	8 (7.4%)	
Age (y)	64.9 ± 10.1	65.5 ± 10.7	0.776
LDL (mmol/L)	2.39 (1.56, 3.06)	2.31 (1.85, 3.17)	0.354
HDL (mmol/L)	1.19 (0.95, 1.36)	1.02 (0.93, 1.26)	0.047
TG (mmol/L)	1.38 (1.05, 1.86)	1.31 (1.01, 1.98)	0.963
TC (mmol/L)	4,45 (3.42, 5.25)	4.32 (3.75, 5.78)	0.524
non-HDL (mmol/L)	3.15 (2.19, 4.06)	3.24 (2.47, 4.61)	0.201
HS-CRP (mg/L)	2.66 (0.86, 5.00)	2.69 (1.20, 5.00)	0.687
BP status			0.080
0	28 (26.7%)	11 (10.5%)	
1	36 (34.3%)	30 (28.5%)	
Systolic BP (mmHg)	132 ± 16	132 ± 19	0.858
Diastolic BP (mmHg)	80 ± 11	81 ± 10	0.664
Smoking			0.080
0	28 (26.7%)	11 (10.5%)	
1	36 (34.3%)	30 (28.5%)	
Diabetes status			0.631
0	45 (42.9%)	27 (25.7%)	
1	19 (18.1%)	14 (13.3%)	
Height (cm)	162.5 (155.0, 168.0)	165.0 (160.0, 168.0)	0.240
Weight (kg)	62.7 ± 10.1	64.0 ± 10.1	0.524
BMI (kg/m^2^)	23.9 ± 2.6	23.9 ± 3.0	0.883
HbA1c (%)	5.90 (5.60, 6.95)	6.15 (5.60, 6.75)	0.585
eGFR (mL/min/1.73 m^2^)	90.85 (75.58, 99.33)	84.90 (63.70, 99.45)	0.231
Apo B (g/L)	0.8 ± 0.3	0.9 ± 0.3	0.039
Lp(a) (mg/L)	141.50 (73.00, 279.75)	254.00 (107.00, 490.00)	0.094
Blood glucose (mmol/L)	6.79 (5.41, 8.20)	5.93 (5.18, 9.41)	0.852
Plaque volume (mm^3^)	47.50 (23.25, 94.00)	74.00 (39.00, 159.50)	0.014
Wall area (mm^2^)	19.20 (15.88, 24.65)	19.10 (16.10, 28.30)	0.452
Median gray scale (dB)	54.50 (41.13, 66.75)	50.00 (40.00, 58.50)	0.404
Maximum area reduction rate (%)	14.00 (8.00, 20.75)	15.00 (11.50, 29.50)	0.061
Normalized wall index	0.32 (0.28, 0.39)	0.35 (0.30, 0.46)	0.070
Plaque thickness (mm)	1.92 (1.58, 2.53)	2.20 (1.64, 2.96)	0.147
Plaque area (mm^2^)	5.95 (3.43, 10.18)	7.80 (4.20, 13.50)	0.179
Lumen area (mm^2^)	38.05 (27.23, 49.93)	34.40 (25.10, 46.15)	0.419
Mean (dB)	28.90 (21.00, 40.13)	24.60 (20.15, 40.40)	0.501
Median (dB)	17.65 (13.53, 20.00)	16.50 (13.15, 19.25)	0.389
Standard deviation (dB)	33.00 (23.25, 46.98)	29.10 (23.05, 48.65)	0.653

LDL, low-density lipoprotein; HDL, high-density lipoprotein; TG, triglycerides; TC, total cholesterol; HS-CRP, hypersensitive C-reactive protein; BP, blood pressure; BMI, body mass index; HbA1c, hemoglobin A1c; eGFR, estimated glomerular filtration rate; Apo B, apolipoprotein B; Lp(a), lipoprotein(a).

**Table 2 diagnostics-12-00256-t002:** Candidate clinical predictors after logistic regression analysis.

Clinical Features	Univariate Analysis		Multivariate Analysis	
	OR (95% CI)	*p* Value	OR (95% CI)	*p* Value
Gender	2.48 (0.98, 6.23)	0.054		
Age	1.01 (0.97, 1.05)	0.774		
LDL	1.32 (0.89, 1.94)	0.166		
HDL	0.22 (0.05, 0.98)	0.047	0.21 (0.04, 0.96)	0.045
TG	1.29 (0.90, 1.84)	0.174		
TC	1.23 (0.92, 1.66)	0.166		
non-HDL	1.34 (0.98, 1.82)	0.062		
HS-CRP	1.02 (0.97, 1.06)	0.443		
BP	0.47 (0.20, 1.10)	0.082		
Systolic BP	1.00 (0.98, 1.03)	0.856		
Diastolic BP	1.01 (0.97, 1.05)	0.661		
Smoking status	0.47 (0.20, 1.10)	0.083		
Diabetes status	0.81 (0.35, 1.88)	0.631		
Height	1.03 (0.98, 1.08)	0.217		
Weight	1.01 (0.97, 1.05)	0.520		
BMI	0.99 (0.86, 1.14)	0.881		
HbA1c	0.95 (0.69, 1.30)	0.741		
eGFR	0.99 (0.97, 1.01)	0.219		
Apo B	4.37 (1.05, 18.19)	0.043	4.60 (1.07, 19.82)	0.041
Lp(a)	1.0012 (0.9997, 1.0027)	0.105		
Blood glucose	1.04 (0.94, 1.15)	0.434		

OR, odds ratio; 95% CI, 95% confidence interval; LDL, low-density lipoprotein; HDL, high-density lipoprotein; TG, triglycerides; TC, total cholesterol; HS-CRP, hypersensitive C-reactive protein; BP, blood pressure; BMI, body mass index; HbA1c, hemoglobin A1c; eGFR, estimated glomerular filtration rate; Apo B, apolipoprotein B; Lp(a), lipoprotein(a)glomerular filtration rate; Apo B, apolipoprotein B; Lp(a), lipoprotein(a).

**Table 3 diagnostics-12-00256-t003:** Candidate predictors of three-dimensional ultrasound of carotid plaque after logistic regression analysis.

Three-Dimensional Ultrasonic Features	Univariate Analysis	
	OR (95% CI)	*p* Value
Plaque volume	1.0046 (0.9998, 1.0093)	0.014
Wall area	1.02 (0.98, 1.06)	0.352
Median gray scale	0.99 (0.98, 1.01)	0.569
Maximum area reduction rate	1.02 (1.00, 1.05)	0.107
Normalized wall index	4.93 (0.25, 97.10)	0.294
Plaque thickness	1.37 (0.87, 2.16)	0.178
Plaque area	1.02 (0.98, 1.07)	0.303
Lumen area	0.99 (0.97, 1.02)	0.530
Mean	1.00 (0.97, 1.02)	0.758
Median	0.98 (0.93, 1.04)	0.507
Standard deviation	1.00 (0.97, 1.02)	0.845

**Table 4 diagnostics-12-00256-t004:** Diagnostic performance of the models.

	Training Set					Verification Set				
	AUC	SEN (%)	SPE (%)	ACC (%)	Y	AUC	SEN (%)	SPE (%)	ACC (%)	Y
Method A	0.648	31.7	84.4	63.8	0.161	0.667	42.9	76.2	62.9	0.191
Method B	0.723	24.4	93.8	66.7	0.182	0.922	78.6	90.5	85.7	0.691
Method C	0.741	41.5	85.9	68.6	0.274	0.939	85.7	85.7	85.7	0.714

SEN, sensitivity; SPE, specificity; ACC, accuracy; Y, Youden index. Method A, Method B, and Method C represent the modeling methods of clinical features (HDL + Apo B), ultrasound features (plaque volume + Rad-score), and the combined model (HDL + Apo B + plaque volume + Rad-score).

## Data Availability

Raw data can be shared from the first author if there is a reasonable request.

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
