# Peer review of "Ultrasound Radiomics Nomogram Integrating Three-Dimensional Features Based on Carotid Plaques to Evaluate Coronary Artery Disease"

_diagnostics, 2022, doi:10.3390/diagnostics12020256_

Round 1
Reviewer 1 Report
Very interesting paper about: Ultrasound Radiomics Nomogram Integrating Three-dimensional Features Based on Carotid Plaques to Evaluate Coronary Artery Disease“ well written with
high scientific content
congratulations
Author Response
First of all,I would like to thank respected reviewer for your patiently review and affirmation of this article.Because of your affirmation, re-checking the revised article is better and the reader can get more valuable information. Thanks again for the help of you sincerely.
Reviewer 2 Report
- the authors should take into account a first attempt from Ciccone MM et al. [Int Heart J. 2011;52(2):72-7] to evaluate the relationship between carotid plaque and their characteristics to the coronary situation. Please discuss such a point.
- pharmacological treatments should be taken into account as they can influence results. THe use of lipid-lowering drug, for instance, is fundamental for the impact on final results.
- how many patients were on atrial fibrillation? AF requires anticoagulation and anticoagulation might impact on vascular walls (see the role of Warfarin in vascular calcification, Scicchitano P et al. Cardiovasc Drugs Ther. 2021 Jun;35(3):505-519.
- I think that a post-hoc sample size calculation should be considered.
Reviewer 3 Report
Authors in this study developed a ultrasound radiomics of carotid plaque features to analyse the severity of coronary artery based on analysis of 105 plaques from 105 patients. Eight ultrasound radiomics features were identified to predict coronary artery stenosis severity with most potential clinical value. In addition, plaque volume of carotid plaques based on 3D US analysis was found to identify patients with intermediate-high SSs. The manuscript is overall well written and easy to follow its contents. It is nicely prepared with section written well. I have some minor comments listed before for improvement:
- Abstract: line 17, spelling error: three-dimensionxiaal-three-dimensional.
- Introduction: page 2, I suggest that authors briefly mention the research gap in the current literature, e.g. what is the status of using ultrasound imaging for prediction of coronary severity in the literature? this will lead to the study purpose smoothly.
- Materials and Methods: fig 2: Fifirstorder features: spelling error again, as it should be Firstorder features.
- Results: well presented.
- Discussion: good discussion with correlation of this study with others in the literature.
- References: why [J] is placed in each reference after the title of study? please double check it.
Reviewer 4 Report
The study is innovative.
Innovative, interesting and useful study for the contribution it can offer in daily clinical practice.
Introduction described well.
Good study design, properly conducted statistical analysis.
Interesting results, especially because they summarize those coming from the new modern radiomics approach and from the conventional laboratory and instrumental exams.
Despite the limitations, however described, the final scientific message remains valid and worthy of attention from the scientific community.
Author Response
First of all,thank you very much for the recognition of this paper.
The author has carefully answered the questions in accordance with the requirements of reviewers and made careful modifications to the article.
Because of your suggestions, the revised article is better and readers can get more valuable information.
Thank dear reviewer again for the help and affirmation of this article.
If dear reviewer has any questions, please feel free to contact me.